# Cost-Efficient Coverage of Wastewater Networks by IoT Monitoring Devices [note 1]

**DOI:** 10.3390/s22186854

**Published:** 2022-09-10

**Authors:** Arkadiusz Sikorski, Fernando Solano Donado, Stanisław Kozdrowski

**Affiliations:** 1Institute of Computer Science, Warsaw University of Technology, Nowowiejska 15/19, 00-665 Warsaw, Poland; 2Institute of Telecommunications, Warsaw University of Technology, Nowowiejska 15/19, 00-665 Warsaw, Poland

**Keywords:** sewer network, wireless sensor network, internet of things, combinatorial optimization, mixed integer programming

## Abstract

Wireless sensor networks are fundamental for technologies related to the Internet of Things. This technology has been constantly evolving in recent times. In this paper, we consider the problem of minimising the cost function of covering a sewer network. The cost function includes the acquisition and installation of electronic components such as sensors, batteries, and the devices on which these components are installed. The problem of sensor coverage in the sewer network or a part of it is presented in the form of a mixed-integer programming model. This method guarantees that we obtain an optimal solution to this problem. A model was proposed that can take into account either only partial or complete coverage of the considered sewer network. The CPLEX solver was used to solve this problem. The study was carried out for a practically relevant network under selected scenarios determined by artificial and realistic datasets.

## 1. Introduction

Wastewater networks are a critical infrastructure: an asset essential for the functioning of society and the economy. Its proper functioning can be impaired by several threats, such as sewage pipe leaks or ruptures, malfunctioning of the wastewater treatment plant (WWTP), etc.

One of the most important threats for its correct functioning in an urban environment relates to the illegal disposal of harsh chemicals in the sewer network. These chemicals may spread beyond the sewer network, and since the capacity of the sewage network and of the WWTP is limited, these chemicals may leak and contaminate groundwater reservoirs, or damage the wastewater treatment plants and render it offline. Examples of unlawful activities of industrial organizations in the sewage network are discharges of: (a) sulfuric acid (H${}_{2}$SO${}_{4}$), resulting from the etching of semiconductors, accumulator acid, or the production of organic chemical substances [1]; (b) sodium hydroxide (NaOH), resulting from cleaning of surfaces in metal processing in industrial applications [2]; (c) sodium sulfate (Na${}_{2}$SO${}_{4}$), resulting from regeneration of cation exchange resins, which are used for softening of water in industrial water treatment [3]. Illegal discharges of such dangerous harsh industrial waste into sewage networks could be harmful for the biological stage of WWTP, its personnel, sewer pipes, and the general public.

Detecting illegal discharges of any of three substances mentioned above can be performed by sampling the wastewater with commercial pH and Electrical Conductivity (EC) sensors. Nevertheless, due to wastewater dilution and mixing effects in sewer pipes throughout the sewage network, the concentration of such substances may be below the minimum detection threshold of such sensors several hundred meters downstream in a populated sub-catchment area. Therefore, it is important to monitor the wastewater composition at multiple points in the sub-catchment area.

As a result, several portable Internet of Things (IoT) systems for monitoring wastewater composition have been proposed in recent years [4,5,6,7,8,9,10,11,12,13,14,15,16,17,18]. These IoT systems are adapted for working at manholes or main sewer lines, and usually comprise a set of sensors (electrochemical sensors, optical sensors, mass spectrometry, ion spectrometry, etc.) for detecting the presence or concentration of specific marker pollutants.

One of such IoT systems is the Micromole system [4,5]. The Micromole system consists of one or more battery-operated devices mounted at sewer main lines. Each device is equipped with pH and Electrical Conductivity (EC) sensors, specially designed for its operation in flowing wastewater [19]. The micromole device is composed of several detachable replaceable modules. In Figure 1 a micromole device comprising five of such modules can be observed. Some of these modules contain batteries, while others contain sensor electronics.

This articles focuses on the planning of an cost-effective positioning of a network of IoT devices monitoring a sewage network. Below we provide an overview of the most recent methods proposed in the literature for the planning of monitoring devices in the sewage network.

This paper is organized as follows. Section 2 presents a description of the most relevant works on the subject. In Section 3, the problem is described and the model is presented with a brief explanation of the dispersion phenomena in wastewater networks. In Section 4 we describe a set of numerical experiments realized within a sewage network in the sub-catchment area of an European city. Section 5 provides the conclusions of our findings.

## 2. Related Work

The SIMONA project [20] has as one of its main goals proposing methods and algorithms for the planning of water quality monitoring stations in sewer systems. Banik et al. propose a set of solutions [21,22,23,24], all of which share the following approach. First, the authors consider as input to the problem a set of time-series of measurements, where one time-series consists of the measurements that would be observed at a given point in the sewage network if one potential source in the network makes a discharge. The measurements provide an indication of the quality of the wastewater, e.g., Electrical Conductivity, following certain given hydraulic conditions. Each measurement of the time-series is then quantized in discrete steps: rounding each measurement to its nearest value in the new scale. As a result, the number of potential different input values is constrained. Next, Banik et al. calculate the information entropy, or information content, of each time-series. After the previously described procedure for pre-processing is executed, Banik et al. consider a dual-objective optimization problem for the placement of the sensor devices. The objective function and meta-heuristic used for finding these solutions vary among Banik et al. contributions, which we summarise below.

In Ref. [21], the two objectives are: (1) maximum information content attained by a group of monitoring stations and (2) minimum the dependency among the monitoring stations. The first objective is achieved by maximizing the joint entropy of the selected monitoring stations, while the second one is attained by minimizing the total correlation of the chosen solution subset of monitoring stations. The set of Pareto optimal solutions is found by using an NSGA-II heuristic. According to Ref. [22], the final decision of selecting the set of monitoring stations from this Pareto front is made by maximizing the amount of information gained by a set of monitors, maintaining the consistency of the selected set of monitors for both variables (concentration and detection time) and having minimum total correlation within a set. The information theory approach taken by Banik et al. has been previously used in related areas [25,26].

In Refs. [22,23] Banik et al. extend their study by considering two additional objectives: detection time of an anomaly and reliability of the solution. The objective related to the detection time aims at minimizing the elapsed time from the discharge event until its detection, when using a fixed number of sensors. The objective related to reliability is related to the number of contamination scenarios that could be potentially correctly detected. Solution to this multi-objective optimization problem were found using the greedy algorithm proposed by Alfonso et al. in Ref. [27], originally designed for other applications.

Our previous work [28] presented the problem of optimising the number of IoT devices in a sewer network, while considering a fixed battery capacity for our sensors in a way that any potential illegal discharge in the sewage network could be detected. In this article we, instead, consider a partial network coverage and include the limitations imposed by sewage physical dimensions on the allocated battery capacities and sensor sampling rates. To the best of our knowledge, this article is the first one in the literature tackling such a problem.

Even though there are design methods in the literature—e.g., Genetic Algorithms [29,30], or Particle Swarm Optimization Algorithms [31]—for solving network coverage problems using Wireless sensor networks (WSN), none of them exploit the flow propagation properties and hydraulic dilution phenomena, as discussed in this article, in their solutions.

## 3. Related Background Knowledge and Proposed Methods

In this manuscript, we consider the problem of optimising the positioning of a wireless sensor network for monitoring the sewer network. In addition, the tackled problem also considers the appropriate allocation of the battery capacity of each sensor device, while considering energy requirements.

Two important requirements for the design of such sensor devices are: (1) to allow its placement in sewer mainline pipes of at least 250 mm of diameter without blocking the flow of sewage, and (2) ease of sensor and battery replacement. Micromole devices fulfil the first requirement by adopting a ring mechanical structure, as shown in Figure 1. Micromole devices fulfil the second requirement by housing electronics into a set of interchangeable modules, each of which share the same dimensions and electronic interconnections. These modules are mechanically and electronically interconnected through the ring mechanical structure, as shown in Figure 1. Since all modules have the same volume, the energy capacity that can be stored using batteries is the same for each module. Nevertheless, even though the energy capacity provided by any module is the same, the number of modules that can be attached to a Micromole device varies and largely depends on the circumference of the ring and, hence, is limited by the pipe diameter where it will be installed: wider pipes allow for a placement of more battery modules for a single device.

The energy consumption of the Micromole device is mostly dependent on the sampling frequency used by its sensors. The sampling frequency shall be set as to avoid situations where the device fails to notice a short discharge, due to its proximity to the source, fast flow speed, or short discharge time. Mitigating such situations can be achieved by assuming that the sampling frequency is dependent on the sewage flow velocity: fast flowing sewage requires high sampling frequency.

In this article we consider that the overall cost of a sensor device comprises the cost of the sensor electrodes themselves—which we consider as a fix cost per sensor device unit—and the cost of the chosen number of allocated battery units.

### 3.1. Pollution Detection and Sensor Localisation

In this article we assume that there is only a single polluting source at a time in the monitored sewer network. This is motivated by the fact that illegal discharges or wastewater pollution is a rare event. Nevertheless, the location of the polluting source, if present, is unknown.

The concentration of an injected pollutant fluctuates from pipe to pipe and, over time, due to the dispersion and dilution effects caused by the mixing of inflows in the sub-catchment area. This effect can be observed in Figure 2, where the EC of wastewater is shown for 82 measuring points downwards a polluting source, from which 50 L of sulphuric acid were disposed.

**Figure 2 sensors-22-06854-f002:**
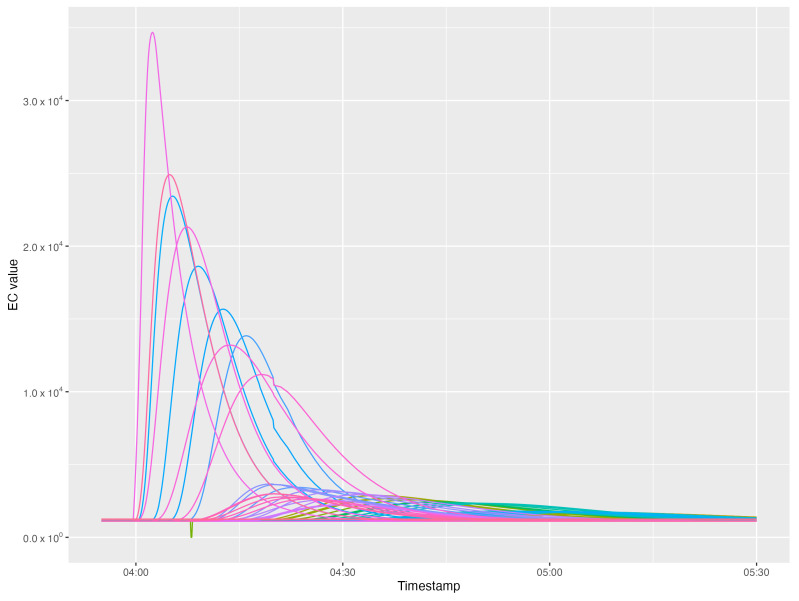
EC broadening and flattening caused by dispersion as seen at different measuring points, when 50 litres of sulphuric acid are discharged 81 manholes upstream from the sink point of the network shown in Figure 3 in low wastewater flow conditions.

A similar effect can be observed when measuring the amount of the diluted pollutant at the same pipe at different points in time during the day: as social and industrial activities demand more usage of water at certain hours, the amount of total flow in a pipe increases and so does the dilution factor of the pollutant. We refer to *flow conditions* as the amount of flow on every pipe at a given point of time.

Due to the dilution effects and limited sensitivity of the sensor devices, the pollutant can only be detected in those pipes where the diluted amount of the substance exceeds the minimum limit of detection of the sensor. We say that a sensor located at pipe *e covers* a potential pollution source $s_{i}$ when considering flow conditions *f*, if the sensor can detect the injection of a pollutant with an anomaly detection method using its collected time-series of sensor measurements. For the purpose of this study, we use a simple threshold criteria as our anomaly detection method: if a measured value exceeds a predefined threshold *Q*, then the sensor can detect the injection of the pollutant. The usage of a simple threshold as an anomaly detection method does not exclude the usage of more complex methods for anomaly detection—such as those based on pattern matching or Artificial Intelligence [32], for instance, or data fusion [33,34].

As a consequence, and given that the flows of wastewater is acyclic in a sewage network, the set of pipes where the pollution from a particular source can be detected form a directed acyclic sub-graph, $G(s_{i})$, of the sewage network. It shall be noted that for two polluting sources $s_{i}$ and $s_{j}$, the corresponding sub-graphs, $G(s_{i})$ and $G(s_{j})$, may have edges in common. If a sensor device is installed in a common edge, it is not possible to discern whether the detected pollutant originates from either $s_{i}$ or $s_{j}$, by only using our threshold criteria.

### 3.2. Model Description

Made assumptions in terms of domain language. These are expressed mathematically in the next sub-section.

  Nodes:a set of nodes denoted as V is defined, each node of this set represents a sewer manhole;a few nodes are distinguished as outlet nodes of the given sewer network;a set $V_{s}\subset V$ is defined and represents nodes that can be sources of undesirable substances.

Edges:a set of directed edges is defined, each edge represents a pipe in the sewer network;any two nodes can have at most only one *direct* connection;edges can be marked as private or public. In Figure 4, dashed lines represent private pipes and solid lines represent public ones;each of these edges is characterized by a parameter that determines the size/flow capacity of water in each pipe;each pipe has a limited cross-area section, which limits the number of slots that can be used for attaching sensors and batteries in a single device. Sensors and batteries can only be installed on a ring device. Such a ring has a fixed cost.

Sensors:sensors detect undesired substances in the sewage, they are to be installed in the edges of the graph;only public edges are eligible for sensor installation while the private ones are not;each sensor has a fixed cost of installation;it is not known a priori how many sensors are required;sensors can only detect the contamination if the concentration in the pipe is not too low, since each sensor has a detection threshold. Each potential source of contamination is associated with a subgraph where the contaminant will be effectively detected and only there it makes sense to install sensors;discharge of undesired substances is a rare event and there can be only one at a time; there is no need to install sensors in a way that several sources can be distinguished;each sensor can sample the sewage at a given frequency—the bigger the flow, the more sensors will be needed to sample the flowing sewage—linearly more (one sensor is enough to sample the sewage having velocity 1 m/s but flow having the velocity 2 m/s requires two sensors).

Battery:sensors require batteries to run;the number of batteries required by each sensor depends on where the sensor will be installed;the number of batteries depends functionally on the sampling frequency, which depends on the flow rate and size of the pipe where the sensor will be installed;each battery has a fixed cost;size of the pipe limits the number of batteries that can be installed in the pipe as well as the number of sensors; the number of sensors and batteries combined is limited by the number of slots on the ring;the installed batteries deplete linearly, all at once.

Coverage of the sources:all potential pollution sources in the sewage network should be covered, i.e., any contamination discharge should be detectable by at least one sensor;one sensor can cover several sources since discharge from only one of them can happen at the time and there is no need to distinguish them;definition of coverage: for each source node $s\in V_{s}$ there is defined a subgraph $G_{s}$ where it makes sense to install sensors. If there is at least one sensor in each such subgraph, we satisfy the coverage condition;any solutions where any pollution source is not covered is not approvable;the coverage constraint is satisfied in Figure 4—the sensor covers both pollution sources that are denoted as red triangles. There is no need to put a sensor in the second leg of the network.

Objective:Minimise the total cost of installing the sensors together with the cost of purchasing batteries for each sensor;We are interested in covering all or parts of the network, so that there is no potential source of contamination that is not detected by at least one sensor.

### 3.3. Mixed Integer Programming Model

The Mixed Integer Programming (MIP) method is proposed to solve the presented problem [35]. The advantage of this method is that it guarantees an optimal solution, as it searches the entire space of admissible solutions to the given problem. In general, the disadvantage of this method is that it often takes a long time to calculate the optimum [36]. In this case, in the problem under consideration, the MIP method performs quite well, even for networks with a large number of nodes.

The following will present the proposed model in mathematical terms. We will describe the definitions of the indices, sets, constants, and variables that appear in this model before the objective function and the necessary constraints are presented. We will operate with the indices *e* and *s*. The former refers to the edges and the latter to the nodes, which are the sources of pollution in the network under consideration. The sets, variables, and constants, on the other hand, are presented in Table 1, Table 2 and Table 3, respectively.

Objective:(1)$$\begin{array}{c} \min\left\{\sum_{e\in E}\left(A\cdot \alpha_{e}+B\cdot \beta_{e}\right)+\Gamma_{e}\sum_{e\in E}\gamma_{e}\right\} \end{array}$$

Constraints: (2)$$\begin{array}{c} \alpha_{e}+\beta_{e}\leq \Lambda_{e}\cdot \gamma_{e}\;\;\;\;\;\;\;\;\;\forall_{e\in E} \end{array}$$(3)$$\begin{array}{c} \sum_{e\in E_{s}}\alpha_{e}\geq \delta_{s}\;\;\;\;\;\;\;\;\;\;\;\;\;\;\;\forall_{s\in V_{s}} \end{array}$$(4)$$\begin{array}{c} \sum_{e\in E_{s}}\gamma_{e}\geq \delta_{s}\;\;\;\;\;\;\;\;\;\;\;\;\;\;\forall_{s\in V_{s}} \end{array}$$(5)$$\begin{array}{c} \beta_{e}\cdot \Theta \geq \Omega_{e}\cdot \Phi_{e}\cdot \alpha_{e}\;\;\;\;\;\forall_{e\in E} \end{array}$$(6)$$\begin{array}{c} \sum_{s\in V_{s}}\delta_{s}\geq \left\lceil \Pi \cdot \right|V_{s}\left|\right\rceil \;\;\;\;\;\;\;\;\;\;\;\;\;\;\; \end{array}$$

Formula (Equation 1) represents the cost function of the presented problem, which is subject to minimisation. Constraint (Equation 2) guarantees us that the number of slots in the ring installed on edge *e* does not exceed the available number of slots. Then, constraint (Equation 3) means that each potential source is covered by at least one sensor. Constraint (Equation 4) tells us that at least one ring must be installed on each edge where the concentration allows detection of harmful substances, while constraint (Equation 5) ensures that the capacity of all batteries must be greater than the lifetime and sampling frequency of the edge *e*. Finally, constraint (Equation 6) indicates the percentage of sources to be covered.

## 4. Experimental Results and Discussion

The proposed mathematical model was tested with two different datasets, each of which was derived from the same sewage network, which is depicted in Figure 3. The sewage network consists of 3297 manholes, 3343 pipes, and 1315 sources of pollution.

The first dataset uses a sub-graph of the base network and consists of 1124 pipes and 402 pollution sources while the second one uses the whole network.

Section 4.1 and Section 4.2 describe how $E_{s}$ sets were created—using discharge simulations and a simplified dispersion model respectively. Section 4.3 describes how sampling frequencies were pre-computed for both datasets. The following two subsections provide results and discussion of the actual cost optimization process using the linear model.

### 4.1. Dataset 1: Simulated Discharges and Dispersion Modelling

All flow and discharge simulations were performed using the software package ++SYSTEM Isar [37], which capabilities were extended by a reaction and transport model based on the concept of total alkalinity in the course of the Micromole project [4].

Due to computational constraints of the ++SYSTEM Isar system, it was not possible to simulate a discharge from every single building in the sub-catchment area. Instead, a subset of 402 buildings were chosen as potential sources of pollution. From every single potential source of pollution, we simulated discharges of 50 L of sulphuric acid, with pH 1 and EC 1400 mS/cm, with low flow conditions and with high flow conditions. Low flow conditions—$f_{L}$—represent the amount of flow found in this sewage network at 03 h 00 m, while high flow conditions—$f_{H}$—represent the amount of flow found in this sewage network at 08 h 00 m during a normal work day.

For establishing the sensor coverage for every particular pipe, we set a threshold for the EC value. In our experiments, we evaluated three different threshold values for EC: $Q_{1}$ = 2 mS/cm, $Q_{2}$ = 3 mS/cm, and $Q_{3}$ = 4 mS/cm, where the normal EC value of wastewater is nearly 1.3 mS/cm. As a result, the combination of the two flow conditions and the three EC threshold values results in six different scenarios that we evaluate below.

### 4.2. Dataset 2: Simplified Dispersion Model

Since discharge simulation is a heavy computational task, an inherited method of proximity generation was introduced to provide test data for a greater number of pollution sources. The algorithm of generating $E_{s}$ sets is presented as Algorithm 1.
**Algorithm 1** Simplified generation of proximities1:**function**generateProximities(G,k)2:    $V_{s}\leftarrow findSourceNodes\left(G\right)$3:    **for**
$s\in V_{s}$
**do**4:          d←findNearestDrainNode(s,G)5:          p←findShortestPathBetween(s,d,G)6:          $E_{s}\leftarrow takeEdgesFromPath\left(p,k\right)$    return $\left\{E_{s}\forall s\in V_{s}\right\}$

The above pseudocode requires some commentary:All source nodes should be found or defined at the beginning; a source node has exactly one outcoming edge and no incoming edges;For each source node *s* the shortest path between *s* and the closest drain node *d* needs to be found. It is the shortest in the terms of lowest number of edges;Each shortest path is shortened and only the first *k* edges are taken. We assume that *k* pipes is enough for a pollutant to become undetectable by a sensor. This simplification is precise enough since pipes in the neighbourhood of each source have comparable lengths. *k* is chosen based on simulated data. We decided to test cases for *k* = 10, 20, 30, 40 since the average and the median length of a path in simulations was about 20 edges.

This method does not require dispersion simulation, which is computationally challenging. Instead, it uses simple graph algorithms, such as shortest path finding. The paths are limited to a length obtained from the simulations run using the smaller network.

### 4.3. Determining Sampling Frequencies for Both Datasets

Sampling frequencies in each pipe had to be calculated for both datasets. The sampling frequency in pipe *e* is affected by two factors:The volume of sewage flowing through the pipe denoted as $u_{e}$. The greater the quantity of sewage in the pipe, the greater sampling frequency needs to be;The area of the pipe’s section, denoted as $\Psi_{e}$, calculated using a standard formula for disk area. The greater the section’s area, the slower the flow in the pipe, so the sampling frequency can be lower.

Assuming that each source *s* continuously adds 1 discrete flow unit of sewage to the network, the flow values are generated as follows ( see Figure 5):For each *e*: set flow value $u_{e}=0$;For each source *s*: find the shortest path between *s* and the closest drain node *d*;For each path *p*: for each edge *e* belonging the path *p*, increase flow value $u_{e}$ by 1 unit.

Finally, sampling frequencies can be determined using the formula $\Phi_{e}=\left(\Phi_{b}+\Phi_{c}u_{e}\right)\cdot \Psi_{e}^{-1}$. $\Phi_{b}$ is the base frequency and $\Phi_{c}$ is the scaling factor of how much sampling frequency needs to be increased per each flow unit.

Values of sampling frequency determined by the described method are presented in Figure 6 as a histogram.

### 4.4. Experiments

This section presents results of numerical experiments obtained with MIP solver and constant parameters presented in Table 4. Our experiments were divided into two cases:Case A—simplified dispersion model data—as explained in Section 4.2—with sampling depending on flow and pipe size;Case B—dispersion model data based on simulated discharges—as explained in Section 4.1—with sampling depending on flow and pipe size.

Each case was tested with Π=0.1,0.2,⋯,0.9,1.0 to determine how the cost changes when the constraint on how many pollution sources have to be covered is changed. The obtained results are presented in Table 5 and in Figure 7 for dataset 1 and in Table 6 and Figure 8 for dataset 2.

Both Figure 7 and Figure 8 show an exponential increase of the cost for an increase in the demanded percentage of the sub-catchment area coverage. For instance, for the scenario when the threshold is set to $Q_{3}$ = 4 mS/cm and there are low flow conditions ($f_{L}$), a reduction of the cost of 47.6%—i.e., from 750 cost units to 393 cost units—can be achieved when relaxing the covered area from 100% to 90%. Similar relative cost reductions can be achieved at 90% coverage for all other five evaluated scenarios in each case.

Such results demonstrate that a wide area coverage is economically feasible for end-users—Law Enforcement Agencies and Environmental Agencies (LEAEA)—interested in monitoring an urban area, if the requirement of covering the whole sub-catchment area is relaxed. From these results, we conjecture that end-users may attempt to select for omission in the planning 10% of sources with a low probability of illegal discharges with the aim of reducing the cost of deployment by almost one half. This conjecture shall be studied in further work.

Figure 9 and Figure 10 show the computational efficiency of the proposed method. Figure 9 shows the time as a function of the percentage coverage of the network for a representative case of the experiment shown in Figure 8. It should be emphasised that the computational time is satisfactory, with the cases between 40% and 80% coverage taking the most computational time.

On the other hand, Figure 10 shows convergence curve as a function of gap and the number of iterations. The gap reflects the difference between the best known bound and the objective value of the best solution produced by a particular algorithm.

Some statistical results concerning space utilization in the edges for both data-set scenarios are also presented in Appendix A.

## 5. Conclusions

This work has addressed the problem of coverage in the sewage network. A model is proposed that provides a coverage problem in a sewer network and at the same time optimises network infrastructure resources such as Micromole rings with modules including sensors and batteries. We proposed the mixed integer programming method, which guarantees to find an optimal solution. In the experiments we used an example of a wide-ranging realistic sewage network from a big-sized city. The method we proposed proved to be effective, giving optimal results in a reasonable computational time.

The convergence curves show an exponential increase in cost for an increase in the desired percentage of coverage of the sub-catchment area. These results show that a wide range of coverage is economically feasible for end users. Based on these results, we conjecture that end-users may try to select up to a dozen percent of sources with low probability of illicit discharges for omission in planning in order to reduce the cost of deployment by almost half. This idea will be the subject of our further research in this area. We plan to develop a model and cost function to locate a potential source of pollutant discharge in the sewer network. We also plan to use evolutionary and bee algorithms if the computation time is long.

## Figures and Tables

**Figure 1 sensors-22-06854-f001:**
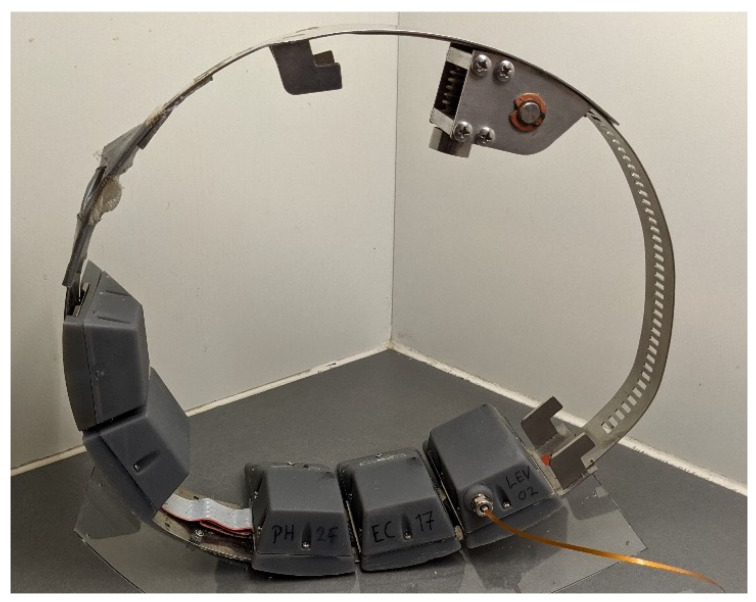
Micromole ring with five modules attached for measuring sewage wastewater physical parameters. From left to right, the attached modules are: battery module, wireless communication module, pH sensor module, Electrical Conductivity sensor module, and a Water Level sensor module.

**Figure 3 sensors-22-06854-f003:**
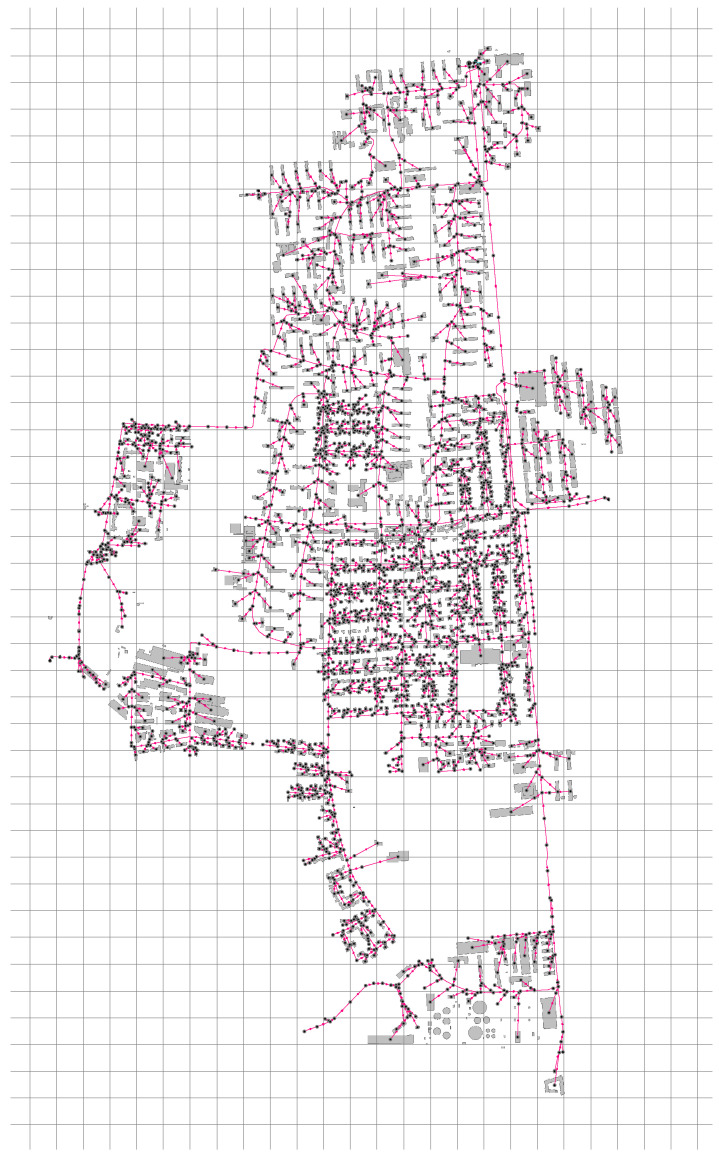
Sewage network used for numerical experiments.

**Figure 4 sensors-22-06854-f004:**
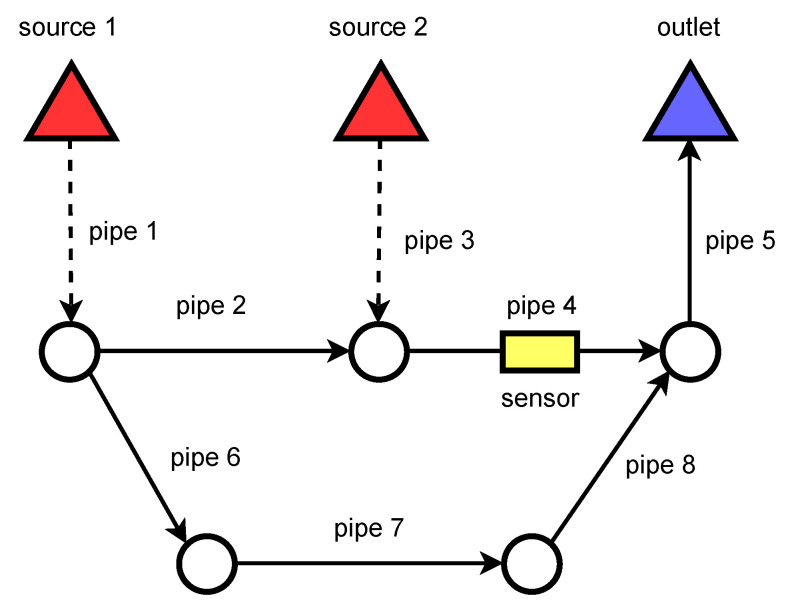
Network coverage. Assuming that $V_{1}=2,4,6,7$ and $V_{2}=4,5$, the sensor installed in pipe 4 is enough to cover both sources. Installing it in pipe 5 would cover only the second source.

**Figure 5 sensors-22-06854-f005:**
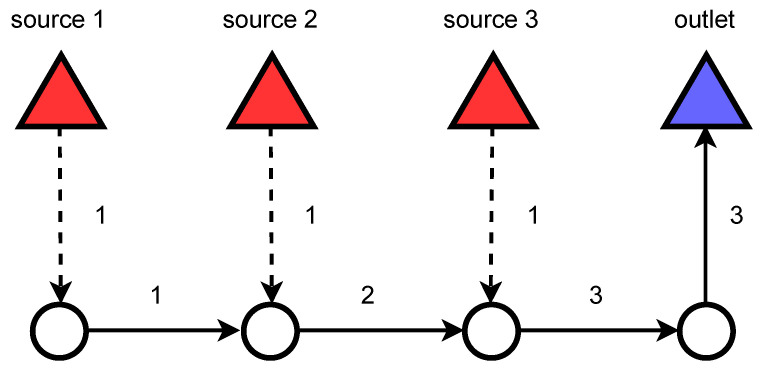
Flow units propagating through the network. The number over the edge is the number of flow units in the pipe. The greater number of flow units next to the outlet node means that a bigger volume of sewage flows in that part of the network when compared to pipes next to the sources.

**Figure 6 sensors-22-06854-f006:**
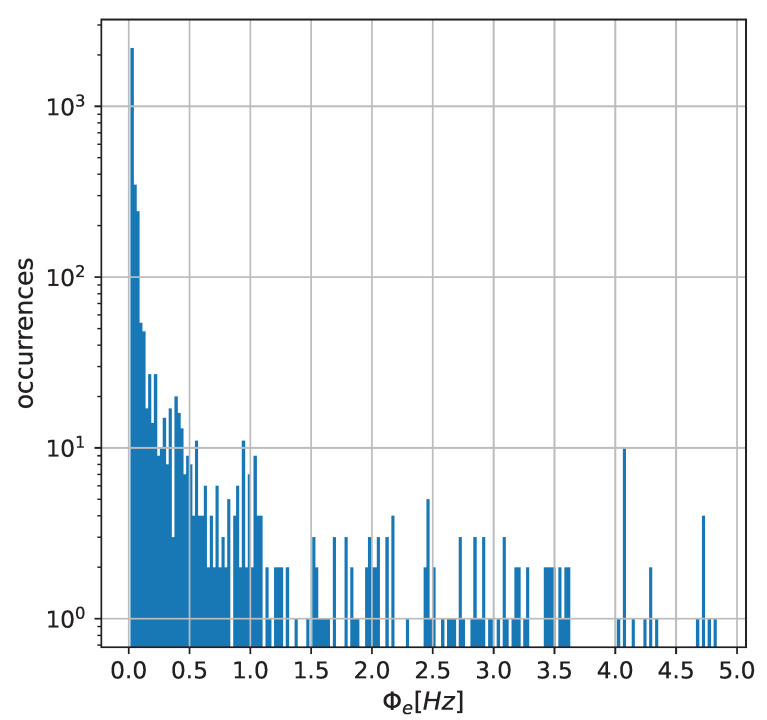
Histogram of sampling frequencies in the network.

**Figure 7 sensors-22-06854-f007:**
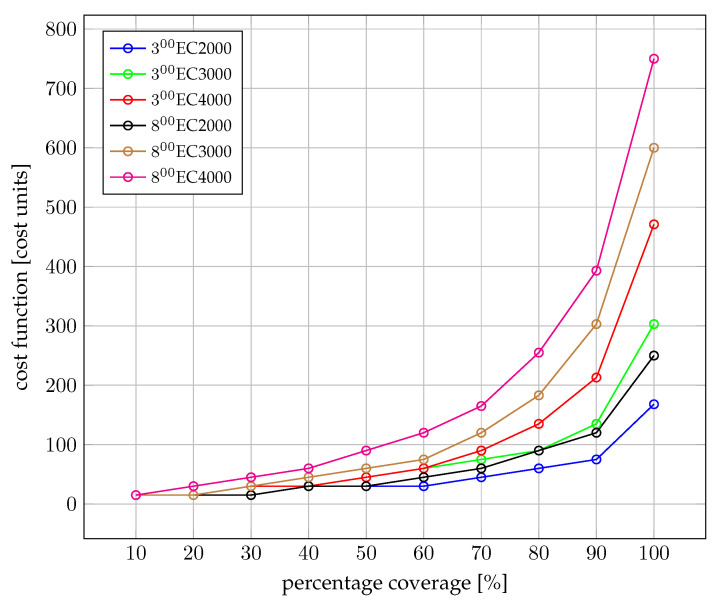
Optimal cost of IoT equipment deployment for dataset 1. Sampling frequency in a given pipe depends on the flow and the size of the pipe.

**Figure 8 sensors-22-06854-f008:**
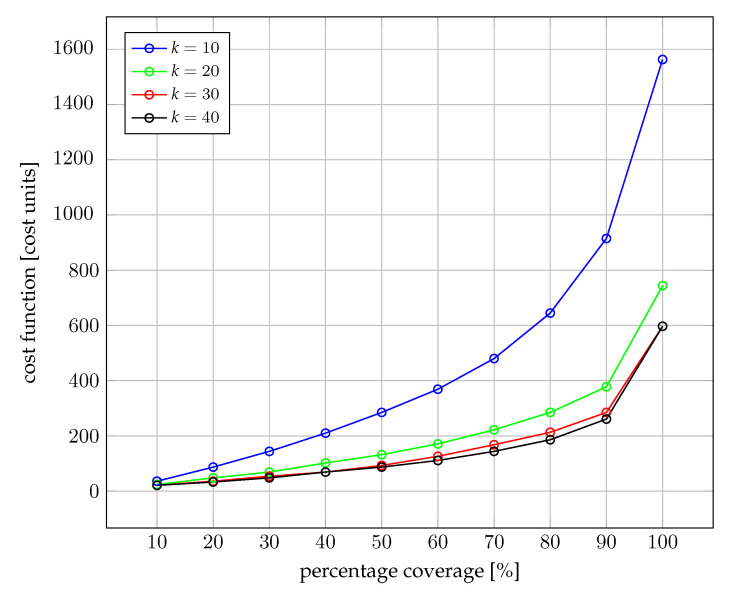
Optimal cost of IoT equipment deployment for dataset 2. Sampling frequency in a given pipe depends on the flow and the size of the pipe.

**Figure 9 sensors-22-06854-f009:**
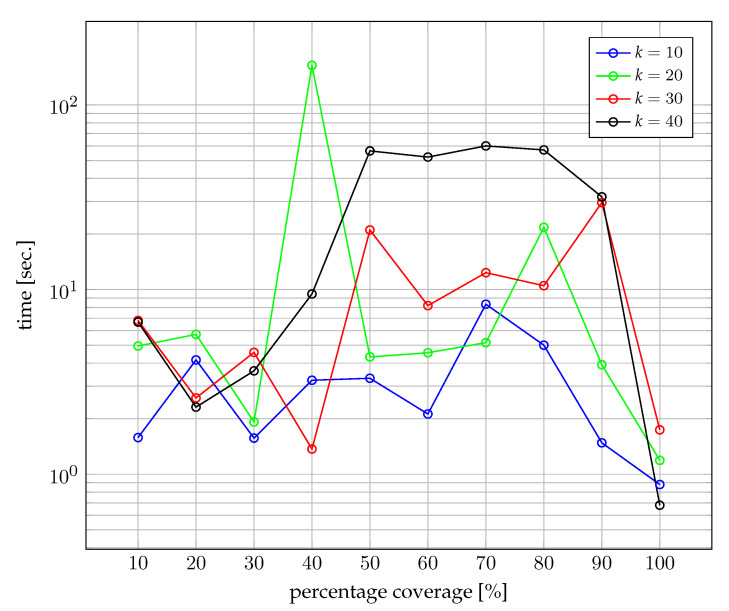
Computation time. Sampling frequency in a given pipe depends on flow and the size of the pipe.

**Figure 10 sensors-22-06854-f010:**
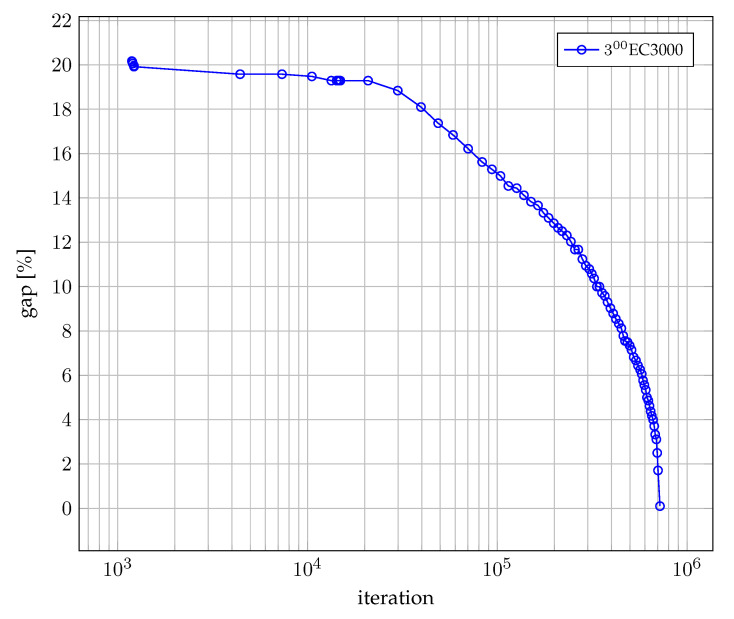
Convergence of the MIP method.

**Table 1 sensors-22-06854-t001:** Sets description.

Set	Description
V	Each vertex s∈V of the graph G represents a manhole
E	The set of directed edges of graph G, which represent the sewage pipes
	over which the sewage flows
$V_{s}$	The set of vertices that could be potential sources of pollution; $V_{s}\subset \;V$
$E_{s}$	The set of edges at which the concentration of pollutants allows
	effective detection of harmful substances after they have been
	emitted from the vertex *s*, also known as proximity; $E_{s}\subset E$.

**Table 2 sensors-22-06854-t002:** Variables description.

Variable	Description
$\alpha_{e}$	$\alpha_{e}\in N$; variable indicating
	how many sensors have been installed at edge e∈E
$\beta_{e}$	$\beta_{e}\in N$; variable indicating
	how many batteries have been installed at edge e∈E
$\gamma_{e}$	Binary; equal 1 if the ring is installed on the edge e∈E;
	0 otherwise
$\delta_{s}$	Binary; equal 1 if the source $s\in V_{s}$ is covered;
	0 otherwise.

**Table 3 sensors-22-06854-t003:** Constants description.

Constant	Description
$\Lambda_{e}$	Number of slots in the ring installed on the edge e∈E
$\Gamma_{e}$	The cost of installing a ring on the edge e∈E
*A*	The cost of one sensor
*B*	The cost of one battery
$\Omega_{e}$	Total battery life at the edge e∈E; expressed in sec;
	We assume that the sensor samples continuously;
	an example value is $10^{6}$ s.
$\Phi_{e}$	Sampling frequency of the sensor at the edge e∈E;
	e.g., once per minute, then $\Phi_{e}=1/60$
Θ	Capacity of one battery; expressed in the number of samples made,
	e.g., $\Theta =10^{5}$, assuming that the batteries are the same
	on each edge e∈E
Π	Percentage of source coverage.

**Table 4 sensors-22-06854-t004:** Values of the used parameters.

Parameter	Value
*A*	7
*B*	3
$\Gamma_{e}$	5
$\Omega_{e}$	$10^{6}$
$\Phi_{b}$	1/60
$\Phi_{c}$	1/60
Θ	$10^{6}$

**Table 5 sensors-22-06854-t005:** Cost function values for the test scenarios of dataset 1.

	Cost [Cost Units]					
Coverage [%]	EC 2000	EC 3000	EC 4000	EC 2000	EC 3000	EC 4000
	3:00	3:00	3:00	8:00	8:00	8:00
10	15	15	15	15	15	15
20	15	15	15	15	15	30
30	15	30	30	15	30	45
40	30	30	30	30	45	60
50	30	45	45	30	60	90
60	30	60	60	45	75	120
70	45	75	90	60	120	165
80	60	90	135	90	183	255
90	75	135	213	120	303	393
100	168	303	471	250	600	750

**Table 6 sensors-22-06854-t006:** Cost function values for the test scenarios of dataset 2.

Coverage [%]	Cost [Cost Units]			
	*k* = 10	*k* = 20	*k* = 30	*k* = 40
10	36	24	21	21
20	87	48	36	33
30	144	69	54	48
40	210	102	69	69
50	285	132	93	87
60	369	171	126	111
70	480	222	168	144
80	645	285	213	186
90	915	378	285	261
100	1563	744	597	597

## Data Availability

Not applicable.

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
