# Peer review of "Cost-Efficient Coverage of Wastewater Networks by IoT Monitoring Devicesâ€"

_sensors, 2022, doi:10.3390/s22186854_

Round 1
Reviewer 1 Report
This article proposes a cost-efficient coverage of wastewater networks by IoT monitoring devices. The problem considered in this article is interesting and important for urban construction. The experiment verifies the performance of the proposed optimization model.
There are some suggestions for this article.
(1) The correlation among the parameters in the mixed integer programming model can be further analyzed to increase the theoretic depth of this article.
(2) The presentation of this article should be carefully checked and revised to be more concise and fluent.
(3) There are some typos in this manuscript, for example 'Such results demonstrate that *a wide are coverage* is economically feasible for end users' (line 328).
Author Response
Dear Sir or Madam,
Please find our revised paper entitled ‘Cost-efficient Coverage of Wastewater Networks by IoT monitoring devices'.
We hope that this paper will find a positive response from you.
With kindest regards,
Stanislaw Kozdrowski

Reviewer 2 Report
1. The division of chapters is not clear enough, and the structure of the article is not properly explained. A reasonable chapter division should be: Introduction, Related Work, Related background knowledge and proposed methods, experimental results and analysis, final discussion, so the context of the article is not clear;
2. The simulation algorithm used in this paper is not accurately described in Section 3.2 about the Simplified Dispersion Model method, which may be inappropriate;
3. The correspondence between dataset 1 and dataset 2 and Fig. 6 and 7 is not accurate;
4. In the discussion of experimental results in Section 4, it is better to describe the comparison of cost function values by table;
5. In the section of experimental results, only the method proposed in this paper has not been compared with the methods already proposed, which may be unconvincing.
Author Response
Dear Sir or Madam,
Please find our revised paper entitled ‘Cost-efficient Coverage of Wastewater Networks by IoT monitoring devices’.
We hope that this paper will find a positive response from you.
With kindest regards,
Stanislaw Kozdrowski

Round 2
Reviewer 2 Report
Most of the previous questions have been modified. Here are some new questions:
1. What is the purpose of finding the acquisition frequency in Section 4.3? And the end of this section does not give the specific value;
2. As for the cost required by all testing equipment, it is not expressed with specific values, but only expressed by cost units, which may not be convincing enough;
3. Figure 6 illustrates the exponential increase in costs as coverage increases in the paper, which is unconvincing just by looking at the figures and tables;
4. The tables in Figure 5 and 6 only show the values in Figure 6 and Figure 7, without calculating the numerical laws and parameter changes that the paper wants to express;
5. In the section of experimental results, only the method proposed in this paper has not been compared with the methods already proposed, which may be unconvincing.
Author Response
Dear Sir or Madam,
We are very grateful again to the reviewer for taking the time to review the
manuscript and providing comments that give us a chance to improve its quality.
Responses to specific comments are provided in the attached pdf file.
Best regards,
Stanislaw Kozdrowski.
